# A Scheduler for Smart Home Appliances Based on a Novel Concept of Tariff Space

**DOI:** 10.3390/s24061875

**Published:** 2024-03-14

**Authors:** Luis Rodolfo Rebouças Coutinho, Giovanni Cordeiro Barroso, Bruno de Athayde Prata

**Affiliations:** 1Department of Electrical Engineering, Federal University of Ceara, Fortaleza 60455-760, CE, Brazil; gcb@fisica.ufc.br; 2Department of Industrial Engineering, Federal University of Ceara, Fortaleza 60455-760, CE, Brazil; baprata@ufc.br

**Keywords:** smart home controllers, load-side management, smart grids, tariff space, load scheduler optimization

## Abstract

The background of this work is related to the scheduling of household appliances, taking into account variations in energy costs during the day from official Brazilian domestic tariffs: constant and white. The white tariff can reach an average price of around 17% lower than the constant, but charges twice its value at peak hours. In addition to cost reduction, we propose a methodology to reduce user discomfort due to time-shifting of controllable devices, presenting a balanced solution through the analytical analysis of a new method referred to as tariff space, derived from white tariff posts. To achieve this goal, we explore the geometric properties of the movement of devices through the tariff space (geometric *locus* of the load), over which we can define a limited region in which the cost of a load under the white tariff will be equal to or less than the constant tariff. As a trial for the efficiency of this new methodology, we collected some benchmarks (such as execution time and memory usage) against a classic multi-objective algorithm (hierarchical) available in the language portfolio in which the project has been executed (the Julia language). As a result, while both methodologies yield similar results, the approach presented in this article demonstrates a significant reduction in processing time and memory usage, which could lead to the future implementation of the solution in a simple, low-cost embedded system like an ARM cortex M.

## 1. Introduction

In 1989, Ref. [1] already pointed to the increasing trend in electric loads quantity and so power demand. Their work discussed many solutions, like the use of dynamic pricing and time-of-use (ToU) tariff, which became a reality in the years to follow. About 20 years later, Ref. [2] listed the same power demand concerns, adding to it a new player: the electric vehicle (EV). The impact of EV on the energy grid was also the main problem for [3,4]. Their research includes scenarios with the coordination of smart chargers. In the same paper, Ref. [2], points to a lack of reliability of the traditional energy grid due to the prospection of renewable energy sources and increased costs to maintain the transmission and distribution networks. Electric energy should be generated closer to its final consumer, and a better communication framework needed to be built, as claimed in [5] when the term smart grid (SG) was used for the first time. Currently, the transport sector is a major source of gas emissions. There are many challenges related to modernizing and increasing the use of public transportation and transitioning from internal combustion to electric vehicles, which cannot be considered gas emission-free if the electric matrix behind it is still based on natural gas or coal [6].

In the context of SG, its evolution is fairly elucidated through the concepts of demand response (DR) and demand-side management (DSM) [7]. The latter reaches out to the end-users through tariff signals offered by the local energy market, often referred to as critical peak pricing (CPP), real-time price (RTP), and ToU [8]. During identified demand peaks, the corresponding hours incur higher charges, thus encouraging users to reschedule their appliance usage to reduce their bills. Legal deals with energy suppliers and smart home controller (SHC) have been important instruments through which customers can optimize their energy consumption behaviors and achieve efficient management of the entire electrical network [9]. A system grid view of the DR problem and their issues related to industrial scenarios can be seen in the reviews by [10,11], respectively.

The Brazilian National Electric Power Agency (ANEEL) classifies electric energy users into two groups: A, which is connected to the grid with voltages higher than 2.3 kV, and B, for voltages below 2.3 kV. Included here are the residential consumers, classified as B1 [12]. For the B1 group, two options of energy tariff are available: conventional tariff ($T_{c}$), which is constant over time, and white tariff, which is subdivided into three hourly constant posts: peak post tariff ($T_{p}$), intermediate post tariff ($T_{i}$), and off-peak post tariff ($T_{f}$). Due to its continental size, the Brazilian energy market is subdivided into several regions, each of which could define their tariff values and post hours in accordance with the previous definitions [13]. Table 1 summarizes the costs of both tariffs in the official currency of Brazil for the local energy market closest to the authors of this paper [14]. Note that there is no billing related to demand response or demand peaks covered by Brazilian official resolution [12] for the B1 group, but this work will consider it due to its relevance for the global scenario.

This work presents a new methodology to schedule home appliances to reduce the energy bill and maintain user comfort when this variable is related to the time shifting of the loads. By exploring some properties of the Brazilian energy tariff, we decomposed the time axis into a geometric space in which the movement of loads through time could be mapped. Analytic analysis of these properties led us to find a specific point in the decomposed time space that ensures relatively lower cost and minimum load shifting. Benchmark trials in the results section show that the proposed methodology is about a thousand times faster than classic algorithms used to solve this class of optimization problem and consumes substantially fewer memory resources, which could lead to a future implementation into a small embedded system.

The remainder of this paper is divided as follows: Section 2 presents the literature review. Section 3 discusses load modeling and defines the study case scenarios. Section 4 states the SHC mathematical model equations and constraints. Section 5 introduces the concept of tariff spaces and sets its properties regarding geometric *locus*. In Section 6, we explain how the proposed methodology works and the simulation results are shown and discussed. Section 7 condenses the contributions of this work and points out some future assignments to improve it.

## 2. Literature Review

Currently, many studies have proposed solutions for energy efficiency in the domestic environment due to the constant increase in both energy consumption and electricity tariffs. In a smart home (SH) scenario, home energy management system (HEMS) controllers are installed to schedule loads at times when the tariff is lower off-peak post [9]. This scheduling typically takes into account the user’s preferences and habits, which can lead to a confrontation with the maximization of the economy.

Considering the scenario with (un)interruptible loads under dynamic pricing, Ref. [15] studied the scheduling problem using the Markov decision process as a possible solver. To provide a strategy for efficient management of electric energy and peak control in a domestic environment, Ref. [16] proposes the design of a SHC using binary linear programming. To deal with uncertainties in appliance use habits and renewable energy generation, Ref. [17] propose a home appliance scheduler combining linear and stochastic programming. Concerned about peak load demand, Ref. [18] modeled the appliances considering the worst-case scenario and photo voltaic (PV) as negative load into CPLEX solver, modeling the scheduling problem as mixed integer programming (MIP). Considering the day-ahead load scenario, a model of a household with PV system and including thermally controlled loads was proposed by [19], which used quadratic programming to minimize the user cost.

Most renewable systems use batteries as energy storage unity, which usage should be modeled and constrained [20]. The same author used two point estimation and gradient-based particle swarm optimization (PSO) to minimize cost and improve demand response in a HEMS. Diesel generators are also a common power source, as considered in [21], which used genetic algorithm (GAs) and linear programming (LP) to model the trade between SHC and local distribution company. Both authors used stochastic models to model dynamic parameters.

The behavior of home appliances is a recurrent concern in this topic due to its unique and intricate characteristics. Subdividing a multiple-stage load into a combination of virtual loads estimated by their peak energy consumption seems to be a reasonable way to handle this problem [22]. Additionally, defining policies based on weather [23] or user life habits [24] are also valid methods to optimize a HEMS.

However, load reallocation can cause discomfort in the user’s habits and trigger physical and psychological issues [25]. Over time, many authors have proposed methodologies to balance the cost versus comfort problem using different techniques like fuzzy logic [25,26,27] integer programming [28,29], convex optimization [30], GAs [9,31,32,33], PSO [34,35], and stochastic programming [36,37,38], to name a few relevant works.

The authors in [39] propose an optimization-based DSM scheduler and energy controller for a smart home considering renewable energy generation and battery storage systems to achieve a reduction in energy cost and peak-to-average ratio in demand and to improve user comfort in terms of thermal, illumination, and appliance usage preference. Their mathematical models are executed in many optimization algorithms.

The scheduling of appliances, considering user habits, can also improve the comfort issue. A Context-Aware Framework, stated on a wireless sensor network to identify behavioral patterns and habits, can generate recommendations that allow energy savings in homes [40]. By monitoring rooms occupancy, a Multi-Agent System can analyze the household data and improve the energy consumption of heating, ventilation, and air conditioning (HVAC) systems [41]. Analyzing patterns from user habits and PV generation a HEMS can avoid power peak consumption penalties [24]. Noninvasive load monitoring approaches and a taxonomy of methodologies to optimize energy consumption have been reviewed by [42].

The studies can be extended to smart builds or even to smart districts by using a two-level approach. The first level is described as the base unit of energy consumption, such as a SH with PV for example. The second level is composed of an array of base units, in addition to shared co-generation and energy storage. For example, in one residential building, each apartment has a solar panel on some windows and share also energy from a PV and/or wind turbine systems on the roof [33,43,44,45].

In preparation for this work, some relevant review articles related to the topic were also found. The authors of [46] identify research trends and patterns in building automation systems, describe sensors and actuators used to build HEMS, and metrics for human comfort evaluation, mainly related to thermal [47] and visual parameters (daylight and glare). The coordination of HEMS due to rebound peaks, instabilities, and contingencies related to the high penetration of these systems in the energy grid is studied by [48], which also lists coordination topologies and mechanisms, as well as implementation prerequisites and mathematical challenges. A comprehensive and in-depth systematic review of artificial intelligence (AI)-based techniques used for building control systems, in terms of human comfort and energy efficiency, has been studied by [49]. A list of papers related to the use of HEMS for different conditions and cases depending upon multiple climate conditions, appliances, controllers with algorithms, distinct home occupants, and their living style has been deliberated by [50], which also identify the main components of HEMS and main optimization techniques to achieve appliance management. The review conducted by [51] determines the primary purposes of smart home systems, listing their key features, characteristics, and requirements, by identifying methods, tools, and technologies to build such systems. By classifying it into residential, commercial, or educational, the authors of [52] provide an overview of the influential factors of energy over-consumption and whether their loads should be directly or indirectly controlled.

## 3. Load Model

During the bibliographic research, we detected that there is no standard for load modeling or classification. However, many authors use similar terms like (non)controllable [9,25,35], (un)interruptible [18,19], and single/multi-period [22,26]. In this paper, loads are classified into two categories, following those stated in [29]:**Controllable load (CL)** encompasses a wide array of devices allowing for manual or remote manipulation. They utilize switches, dials, or digital interfaces to adjust operations. Integrated into SH ecosystems, users can oversee appliances using smartphones or voice assistants, enhancing convenience and energy efficiency. Examples of CL are air conditioners, pool filter pumps, non-programmable washing machines, dishwashers, irons, or even outdoor lighting.**Detectable load (DL)** refers to an electrical device or equipment that can be identified and monitored within a SH ecosystem. Unlike CL, detectable ones are not typically designed for remote manipulation or control. However, we can estimate their energy consumption by comparing the energy measurements of the smart meter (SM) and all other devices connected to a HEMS. Examples of DL are audiovisual equipment, personal computer systems, indoor lighting, toasters, refrigerators, and freezers.

The parameters of the *i*th CL in a set, which were used to structure the programmer model and simulations, are presented in Table 2 and are closely related to scheduling problem modeling [53].

Figure 1 provides an overview of the load parameters listed in Table 2, illustrating their positioning over time for a generic or randomly drawn load with multiple discreet power stages (gray object).

On the left side of Figure 1, the release $L_{i}.r$ and expected activation time $L_{i}.e$ instants are shown. On its right side, the deadline $L_{i}.d$ and the range of power over time $L_{i}.P\left(t\right)$ are depicted. Above the gray area, the load length $L_{i}.W$ is shown. Below it, the start $L_{i}.s$ and finishing time $L_{i}.f$ instants are marked. Details of the code that realizes this structure in Julia language can be found in the link provided in Appendix A.

In this work, we also considered that a complex or multistage load could be simplified as a combination of single small loads [22]. Figure 2 demonstrates this process for a two-stage load $L_{i}$.

In this example, we split its duration into stages, each with its own start and finish times. We then adjusted the parameters for the release ($L_{b}.r$) and expected time ($L_{b}.e$) of the second stage to align with the deadline ($L_{a}.d$) of the first stage. Moreover, the first stage inherits the release ($L_{a}.r$ = $L_{i}.r$) and expected time ($L_{a}.e$ = $L_{i}.e$) from the original load $L_{i}$, while the second stage inherits its deadline ($L_{b}.d$ = $L_{i}.d$).

The simulation step or sampling rate is also an important variable and should be considered as minimum as possible to achieve flexibility in scheduling [23]. All simulations and benchmark results were obtained using Δt=5 min. However, for some later illustrations, it will be stated as Δt=30 min for better graphical comprehension.

### Simulation Scenarios

Nine simulation scenarios are proposed in this work. The first one is related to a real house described in [29] and also studied in [9,25,35]. The details of appliances are described in Table 3. This set of loads has been considered due to the known results by the authors of this paper from previous works, serving as a compass to ensure the methodology presented in this paper.

The last eight sets of loads were randomly generated and utilized to collect benchmarks for execution time and memory usage, corresponding to load quantities of 10, 25, 50, 75, 100, 250, 500, and 750. These specific load quantities were chosen to assess the impact of increasing load set sizes on performance parameters. All benchmark output details are described in a GitHub link in Appendix A.

The common attributes across all simulation scenarios are as follows: (a) The sampling interval, denoted as Δt, is fixed at 5 min; (b) each ensemble of ten loads adheres to a daily demand threshold of 4.0 kW. This threshold is depicted by an inverted Gaussian distribution centered at 18:30 h, with a negative amplitude of 25%, serving to simulate a reduction in the demand threshold to accommodate the DL; (c) the Brazilian ToU tariffs: constant and white.

For each load in any random scenario, the restrictions from Equations (Equation 1) and (Equation 2) were applied. Additionally, the relevance factor μ previously defined in Table 2 has been set to one to avoid any attenuation on the evaluation process of the comfort goal.
(1)$$L_{i}.W\leq 6\;h$$
(2)$$L_{i}.r\leq L_{i}.e.,\leq L_{i}.d-L_{i}.W$$

For each controllable load $L_{i}$ in a SH context, the threshold in Equation (Equation 1) has been established based on the authors’ common understanding that a controllable appliance would rarely operate for more than 6 h. This time length is represented by the variable $L_{i}.W$. Nevertheless, this value could have been set to any appropriate value. The constraints outlined in Equation (Equation 2) specify that the expected activation time ($L_{i}.e$) for a load must fall within the release time ($L_{i}.r$) and its deadline($L_{i}.d$), adjusted by the duration of the load ($L_{i}.W$). All these variables have been previously defined in Table 2.

## 4. SHC Classic Model

In this paper, we assume that a SHC is connected to all controllable loads and is capable of performing their scheduling. For this control to be possible, the SHC must receive information about energy billing, white tariff ($C_{w}\left[t\right]\in \left[T_{f},T_{i},T_{p}\right]$) and constant tariff ($T_{C}$), controllable residential loads set ($L_{m}$), residential load activation preferences ($L_{m}.e$, $L_{m}.r$, $L_{m}.d$), and comfort level ($L_{m}.\mu$). To achieve this goal, we model the data related to residential loads, including the consumption profile ($f_{1}$) and the residential comfort profile ($f_{2}$). This modeling process enables us to understand consumption patterns, identify potential savings, and optimize comfort levels through a day-ahead load schedule.

### 4.1. Cost Model—f1

The mathematical definitions of residential load at the grid level employed in this paper are akin to those presented in prior works [16,29,43]. The mathematical model of residential loads corresponds to Equation (Equation 3), which incorporates the following premises: *M* schedulable loads, *N* daily samples, a sampling interval Δt. All these variables follow the notation described in the Load Model Section.
(3)$$f_{Fcost}=\sum_{i=1}^{M}\sum_{j=L_{i}.s}^{L_{i}.s+L_{i}.W}\left({\bar{P}}_{i}\left[j\right]\frac{\Delta t}{60}C_{w}\left[j\right]\right)$$
subject to the following constraints:(4)$$L_{m}.r\leq L_{m}.s\leq L_{m}.d-L_{m}.W$$
(5)$$\sum_{j=1}^{N}\left(\sum_{i=1}^{M}{\hat{P}}_{i}\left[j\right]\right)\leq P_{j}$$
where ${\bar{P}}_{i}$ and ${\hat{P}}_{i}$ are, respectively, the average power and the peak power of the *i*th L load, and $P_{j}$ is the maximum demand restriction.

The limitations outlined in Equation (Equation 4) specify that the timing of activation for the *i*th load must fall within the user-defined release and deadline time instants, in the same terms Equation (Equation 2) was defined. Additionally, the loads must not surpass the threshold demand ($P_{j}$) at the *j*th activation time, as indicated by the constraints presented in Equation (Equation 5).

The cost function ($f_{1}$) defines the economic savings due to SHC normalized by the cost in constant tariff. The first and second terms in Equation (Equation 6) correspond to the costs resulting from the user preference profile and the SHC scheduling, respectively.

The normalized economic savings, denoted by the cost function ($f_{1}$), articulate the financial benefits of dynamic tariff attributed to the SHC, normalized against the costs in a constant tariff setting. The initial and subsequent elements in the numerator of Equation (Equation 6) represent the costs associated with the user preference profile and the scheduling facilitated by the SHC, respectively, considering ToU white tariff.
(6)$$f_{1}=\frac{\sum_{i=1}^{M}\left(\sum_{j=L_{i}.e}^{L_{i}.e+L_{i}.W}\left({\bar{P}}_{i}\left[j\right]\frac{\Delta t}{60}C_{w}\left[j\right]\right)-\sum_{j=L_{i}.s}^{L_{i}.s+L_{i}.W}\left({\bar{P}}_{i}\left[j\right]\frac{\Delta t}{60}C_{w}\left[j\right]\right)\right)}{\sum_{i=1}^{M}\sum_{j=L_{i}.s}^{L_{i}.s+L_{i}.W}\left({\bar{P}}_{i}\left[j\right]\frac{\Delta t}{60}T_{C}\right)}$$

In this context, $f_{1}\geq 0$ ensures that the schedule proposed by the SHC is deemed acceptable by the algorithm as a valid solution for the user.

### 4.2. Comfort Model—f2

The comfort model, adapted from [29,43], takes into account the comfort relevance level of a load *i* as a measure of how much it deviates from the expected activation time by the user. To facilitate this, users are required to register residential loads eligible for scheduling in the SHC, along with specifying comfort relevance values ($0\leq L_{i}.\mu \leq 1$) and the load activation parameters in terms of release ($L_{i}.r$), deadline ($L_{i}.d$), and expected ($L_{i}.e$) time instants.

Equation (Equation 7) delineates the comfort function. The initial term signifies the activation window of a load *i* concerning the user’s preferences, serving as a benchmark for computing normalized comfort. The subsequent term quantifies the discrepancy between the time instant ($L_{i}.s$) chosen by the SHC and the user’s preferred time ($L_{i}.e$). This difference is adjusted by the comfort relevance ($L_{i}.\mu$) associated with the *i*th load. Note that, if μ is equal to zero, for a certain load, it does not matter when this load is scheduled, as the comfort related to the load is set to maximum value; it is equal to one.
(7)$$f_{2}=\frac{\left[max(\left|L_{i}.r-L_{i}.e.,\right|,\left|\left(L_{i}.d-L_{i}.W\right)-L_{i}.e.,\right|)\right]-L_{i}.\mu \left|L_{i}.s-L_{i}.e\right|}{\left[max(\left|L_{i}.r-L_{i}.e.,\right|,\left|\left(L_{i}.d-L_{i}.W\right)-L_{i}.e.,\right|)\right]}$$

For a specific load *i* with a comfort relevance of $L_{i}.\mu =1$, this parameter attains its highest value when the scheduled time by the SHC aligns closely with the user’s preferred time ($L_{i}.s\approx L_{i}.e$). However, if $L_{i}.s\approx L_{i}.r$ or $L_{i}.s\approx \left(L_{i}.d-L_{i}.W\right)$ (at the opposite end of the load activation window), the comfort level will be minimal. This occurs because the operation cycle commences at a time furthest from the one designated by the user as the preferred time.

### 4.3. JuMP and Hierarchical Algorithm

Julia Modeling Language for mathematical optimization (JuMP) is a modeling language [54] that condenses a collection of supporting libraries and packages running in Julia language [55] that makes it prone to formulate and solve different problem classes related to optimization. The Multi-Objective Algorithms package [56] provides many classic implementations ready to use. The best benchmark results were achieved with the hierarchical algorithm.

The hierarchical multi-objective algorithm organizes its approach to return a single point via an iterative scheme. First, it partitions the objectives into sets according to the objective priority. Then, in descending order of priority, it formulates a single-objective problem by scalarizing all objectives with equal weights. Next, it constrains these objectives to be at most relative tolerance worse than optimal in future solves.

In other words, it solves the model up to a given MIP gap to obtain an optimal value for the first objective function. Then, given the model restrictions, it optimizes the second objective using the first set of values to constrain the feasible set of the next optimization, such that the evaluated solution cannot become worse without first taking into account some predefined tolerance.

Finally, it proceeds to the next set of prioritized objectives. The solution represents a single point that trades off the various objectives. To save memory space, the implementation of this algorithm in JuMP development framework does not record the partial solutions found along the way [54,57,58]. All code related to the implementation of SHC Classic Model can be found in the link provided in Appendix A.

## 5. Tariff Spaces

The reader is now invited to look at Figure 3. The squared chart represents a zoom-around peak post of white tariff over time. The white outer regions are related to off-peak post, while the two thin yellow regions are related to the intermediate post and the central red area represents the peak post. Each square along the chart represents a sample period, which, for didactic purposes, has been set to 30 min. The long blue rectangles represent a generic four-hour-long load started in three different instants.

Note that while the four-hour-long load crosses the tariff posts, it is possible to evaluate how many discrete samples fit in each time post. These quantities are shown on the right side of each load representation in Figure 3.

From now, we assume that each time (or region) post in white tariff could be modeled as an independent dimension so that we could create a three-dimensional vector with components ($\hat{f},\hat{i},\hat{p}$), respectively, to white tariff off-peak post, intermediate post, and peak post, whose lengths are the load length portion that fits inside of each post region. This vector space is named here as **Tariff Space**.

The process for a load *L*, which quantifies how many discrete samples will fit in each time post according to load length (L.W, L.WΔ) and its start time (L.s), is defined as **time decomposition into tariff space**. Its output is a vector in Tariff Space, as shown on the right side of Figure 3.

The code related to time decomposition into tariff space and its reverse operation can be found in the link provided in Appendix A. As the vector resulting from time decomposition has been stated, we now can evaluate the cost of a load into a white tariff scenario using dot product:(8)$$C_{w}=k\cdot \bar{P}\cdot \left[\left(\hat{f},\hat{i},\hat{p}\right)\cdot \left(T_{f},T_{i},T_{p}\right)\right]$$
where:(9)$$|\hat{f}\left|+\right|\hat{i}\left|+\right|\hat{p}|=L.W\Delta ;$$
(10)$$k=\left(\frac{60}{\Delta t}\right);$$

$\bar{P}$ is the average power of a load and *k* is the discrete amount of time related to one hour due to sample rate Δt. All these symbols were defined in Table 2. The values in vector $\left(T_{f},T_{i},T_{p}\right)$ represent the white tariff post costs, as previously stated in Table 1.

The minimum, maximum, and normalized costs of a load can also be written as:(11)$$C_{min}=k\cdot \bar{P}\cdot L.W\Delta \cdot T_{f}$$
(12)$$C_{max}=k\cdot \bar{P}\cdot L.W\Delta \cdot T_{c}$$
(13)$$C_{norm}=\left(\frac{C_{w}}{C_{max}}\right)$$

Note that the maximum is a relative value and is evaluated using constant tariff value because our goal is to reduce the bill relative to this reference value.
(14)$$C_{min}\leq C_{w}\leq C_{max}⟷L.W\cdot \Delta T_{f}\leq \left[\hat{f}\cdot T_{f}+\hat{i}\cdot T_{i}+\hat{p}\cdot T_{p}\right]\leq L.W\cdot \Delta T_{c}$$

As cost margins have been defined, we can analyze the extreme points of Equation (Equation 14). Solving the equality at the lower bound yields the expression seen in (Equation 15). This indicates that to achieve this threshold, the total load length should fall within the off-peak post or, in other words, the point (L.WΔ,0,0). Solving the equality at the upper bound yields the two points expressed in (Equation 16). These three points delimit a region into tariff space in which the cost of a load in the white tariff scenario is less or equal to the constant tariff.

Furthermore, note that Equation (Equation 9) is an equilateral triangle that encompasses all possibilities of combinations for ($\hat{f},\hat{i},\hat{p}$) limited to L.WΔ. All these regions, along the points related to a one-hour-long load crossing the tariff regions, can be seen in Figure 4. The points related to the load movement have been colored according to their normalized cost, so the reader can see how their price change over the gray triangle plane surface. The region delimited by a red triangle represents the region of lower cost.
(15)$$\hat{i}=-\hat{p}\frac{T_{p}-T_{f}}{T_{i}-T_{f}}<0$$
(16)$$\left\{\begin{array}{ccc} if\;\hat{i}=0 & \hat{p}=L.W\Delta \frac{T_{c}-T_{f}}{T_{p}-T_{f}} & \hat{f}=L.W\Delta -\hat{p}=L.W\Delta \frac{T_{p}-T_{c}}{T_{p}-T_{f}}\\ if\;\hat{p}=0 & \hat{i}=L.W\Delta \frac{T_{c}-T_{f}}{T_{i}-T_{f}} & \hat{f}=L.W\Delta -\hat{i}=L.W\Delta \frac{T_{i}-T_{c}}{T_{i}-T_{f}} \end{array}\right.$$

Observe that the one-hour load "walks" through the side of the triangle only. This occurs because time decomposition for this load would never have three components as its length fits entirely into all three tariff regions or between its adjacent transitions in pairs. For loads with length less than or equal to L.Δt only the vertices of the triangle should be considered.

The next relevant load movement graphic is shown in Figure 5 and represents a six-hour-long load. Note that the behavior in tariff space is quite different from the observed in Figure 4. This behavior can easily be modeled accordingly only to the load length. Those patterns, called here geometric *locus* of a load, are shown in the six equations that follow. More examples of load decomposition into time space can be found in the link provided in Appendix A.
(17)(L.WΔ,0,0);(0,L.WΔ,0);(0,0,L.WΔ)
(18)$$\hat{i}=L.W\Delta -\hat{p}\leq k;\hat{f}=0$$
(19)$$\hat{i}=L.W\Delta -\hat{f}\leq k;\hat{p}=0$$
(20)$$\hat{i}=k,\hat{f}=L.W\Delta -\hat{p}-k;$$
(21)$$\hat{i}=\left(L.W\Delta -3\cdot k\right)-\hat{f};\hat{p}=3k$$
(22)$$\hat{i}=2k;\hat{p}=3k;\hat{f}=L.W\Delta -5\cdot k$$

The points shown in Equation (Equation 17) are related to a load whose length is less or equal to Δt, as mentioned before. Equation (Equation 18) represents a triangle side that connects axis $\hat{p}$ to $\hat{i}$ and signifies a linear trade-off for a load whose length is lower or equal than an hour. A similar case occurs in Equation (Equation 19) that represents the triangle side that connects axis $\hat{f}$ to $\hat{i}$.

Equation (Equation 20) is applied when the load length is greater than an hour but less or equal to four hours. It represents a parallel line to the triangle side that connects the axis $\hat{f}$ to $\hat{p}$. Note that, while a load longer than an hour crosses the intermediate post, that component remains constant and equal to an hour in size (represented by variable *k*, defined in Table 2). Additionally, it is important to recognize that this triangle side could never be reached as we could not split a single load into two parts. As stated in the load model section, multiple-stage loads should be modeled as an array of single, indivisible loads with a shared deadline and release time.

The last two Equations (Equation 21) and (Equation 22), are related to loads whose lengths are greater than 4 h. It is important to note that for loads with a length of 4 h or longer, as they cross the intermediate and peak posts their respective components should remain constant and equal to the regions occupied. Equation (Equation 21) models a line parallel to a side that connects axis $\hat{f}$ to $\hat{i}$. Finally, Equation (Equation 22) is a single point into tariff space that exists while the load is placed through the three-time posts and is also larger than both intermediate and peak posts.

## 6. Proposed Methodology

Once we have defined all possible geometric *loci* for an appliance, it is pretty visible that only the lines defined by Equations (Equation 19) and (Equation 20) could reach the lower cost region. The analysis of upper bounds in Equation (Equation 14) gives us two points that could be combined to generate a parametric line Equation (Equation 23). The intersection between this line and load geometric *locus* will give us the solution to our schedule problem,
(23)$$\left\{\begin{array}{c} \hat{f}=L.W\cdot \frac{t_{p}-t_{c}}{t_{p}-t_{f}}+\lambda \cdot \left(L.W\cdot \frac{\left(t_{p}-t_{i}\right)\cdot \left(t_{c}-t_{f}\right)}{\left(t_{p}-t_{f}\right)\cdot \left(t_{i}-t_{f}\right)}\right)\\ \hat{i}=\lambda \cdot L.W\cdot \frac{t_{c}-t_{f}}{t_{i}-t_{f}}\\ \hat{p}=L.W\cdot \frac{t_{c}-t_{f}}{t_{p}-t_{f}}\cdot \left(1-\lambda \right) \end{array}\right.$$
where λ is the parametric variable for Equation (Equation 23)

Equations (Equation 19) and (Equation 20) can also be rewritten in parametric form,
(24)$$\left\{\begin{array}{c} \hat{f}=L.W\cdot \left(1-\alpha \right)\\ \hat{i}=\alpha \cdot L.W\\ \hat{p}=0 \end{array}\right.$$
(25)$$\left\{\begin{array}{c} \hat{f}=\rho \cdot \left(L.W-k\right)\\ \hat{i}=k\\ \hat{p}=\left(L.W-k\right)\cdot \left(1-\rho \right) \end{array}\right.$$
where α and ρ are parametric variables for Equations (Equation 24) and (Equation 25), respectively.

Evaluating the interception point between Equations (Equation 23) and (Equation 24), we find a point described in Equation (Equation 26), which is one of the points that belong to ones listed in Equation (Equation 16). Graphically, it is indeed the point at lower cost region border where the cost of a load in white tariff is equal to the cost in constant one. As we need a relatively lower cost, we could use the *rounding floor function* to reach the next point inside the triangle. Note that by choosing this first inner point, a load whose expected time is in intermediate or peak post has minimum movement through time space, that way both objectives, cost, and comfort (as defined in Section 4.2 and Equation (Equation 7)) are achieved.
(26)$$P_{best1}\;=\;\left(\hat{i}=\left\lfloor L.W\cdot \frac{T_{i}-T_{c}}{T_{i}-T_{f}}\right\rfloor ,\;\hat{f}=L.W-\hat{i},\;\hat{p}=0\right)$$

Calculating the interception point between Equations (Equation 23) and (Equation 25), we find the point described in (Equation 27).
(27)$$P_{best2}\;=\;\left(\hat{p}=\left\lfloor L.W\cdot \frac{k\cdot \left(T_{f}-T_{i}\right)-L.W\cdot \left(T_{f}-T_{c}\right)}{T_{p}-T_{f}}\right\rfloor ,\;\hat{i}=k,\;\hat{f}=L.W-k-\hat{p}\right)$$

The criteria to choose between $P_{best1}$ or $P_{best2}$ depend on the load length relative to *k* and the sign of $\hat{f}$ component. The flowchart in Figure 6 shows the decision process between the two values. Appendix A has a link to all code for the geometric search process (GeoFind for short).

### 6.1. Analysis of a Load Fully into Off-Peak Post

As stated before, the methodology previously discussed is applicable only if the load has its expected start time inside the interval composed of the intermediate and peak posts. If a load is scheduled by a user fully into the off-peak post, no movement should be made with it, as it is already with maximum comfort (see Section 4.2 for details) and lower possible cost.

### 6.2. Analysis over Defective Geometric Locus

Due to restrictions caused by the release or deadline instants, some loads may have not the full capability of moving through their geometric *locus*. As a result, such loads may be considered defective. To illustrate, consider an example of a load whose data could be read in Table 4 and its geometric *locus* seen in Figure 7. Note there is no intersection between the geometric *locus* and the lower cost region. In that specific case, we should look at the edges of the possible start times to find the schedule position with higher component $\hat{f}$, and to the expected schedule time, then evaluate the ratio between cost, Equation (Equation 8), and comfort, Equation (Equation 7), for this three instants, as shown in Equation (Equation 28).
(28)$$\begin{array}{c} \left\{\begin{array}{c} R1=Comfort(L,L.s[1])/Cost(L,L.s[1])\\ Rf=Comfort(L,L.s[end])/Cost(L,L.s[end])\\ Re=Comfort(L,L.e)/Cost(L,L.e) \end{array}\right.\\ \;\;\;\;\Rightarrow R1\geq Rf\;?\;(R1\geq Re\;?\;L.s[end]:L.e)\;:\;(Rf\geq Re\;?\;L.s[end]:L.e) \end{array}$$

The best ratio result between comfort and cost should be returned as a solution in this case. A short implementation based on the statements of this and the previous subsection are shown in Algorithm 1.
**Algorithm** **1** Best_Geofind Algorithm.**Require:** 
L::BasicLoad  $Pe\leftarrow Decompose_{T}ime\left(L,L.e\right)$  **if** 
Pe[1]=L.CΔ 
**then**        **return** L.e  **end if**  
A←GeoFind(L)  $t\leftarrow Recompose_{T}ime\left(A\right)$  **if** (L.s[1]≤t) and (t≤L.s[end]) **then**        **return** t  **end if**  $C1\leftarrow Eval_{B}asicLoad_{C}omfort\left(L,L.s\left[1\right]\right)/Eval_{B}asicLoad_{C}ost\left(L,L.s\left[1\right]\right)$  
$Ce\leftarrow Eval_{B}asicLoad_{C}omfort\left(L,L.e\right)/Eval_{B}asicLoad_{C}ost\left(L,L.e\right)$  
$Cend\leftarrow Eval_{B}asicLoad_{C}omfort\left(L,L.s\left[1\right]\right)/Eval_{B}asicLoad_{C}ost\left(L,L.s\left[end\right]\right)$  
val,tempo←C1≥Cend?(C1,L.s[1]):(Cend,L.s[end]) 
**return** 
Ce≥val?L.e:tempo

### 6.3. Analysis of Power Demand Response

To date, all analyses that have been made have focused on only one load. As loads could be considered independent, a feasible solution for an array of loads would be to iterate the geometric search through a loop and return all schedule instants by recomposing time after finding the points in tariff space. One side effect of this solution is that we cannot yet add any constraint about demand peaks or maximum power over time.

To cover this issue, a few hypotheses have been formulated and two of them tested through simulation. Both use the hierarchical algorithm in combination with the output results of geometric search to try to speed up their execution and reduce memory use. The first trial consists of initializing the hierarchical algorithm with instant values evaluated with the geometric search.

The second approach iterates through the scheduled loads and locates the ones whose summed power exceeds the demand restriction. After identifying the loads that are causing the surge peak, they are cut from the original problem set and passed as parameters to the hierarchical algorithm, which will attempt to reschedule them within 95% or 90% of the original constraint. This reduction is needed to avoid another demand peak by reinserting the loads into the full set. Then, we check if all loads fit into the demand constraint. In the negative case, another cut is made.

Once all loads fit within the demand constraint, the iterations end. This last methodology has been named the hybrid algorithm. A short version of this method can be seen in Algorithm 2, and its full codification in Julia language can be found in the link in Appendix A. Functions that iterate through a set of loads receive the suffix *Vector*. As presented in Algorithm 1, the Best_Geofind function combines Geometric search with statements in Section 6.1 and Section 6.2.
**Algorithm** **2** Hybrid Algorithm.**Require:** 
H::VectorBasicLoad**Require:** 
demandScale::Float64           ▹ default value 0.95**Require:** 
maxIterations::Int64             ▹ default value 5  t←Best_Geofind_Vector(H)  count←0  **while** true **do**       Hcut,B←find_Demand_Peaks(H,t)    ▹ return loads and its indexes       **if** isempty(B) or count ≥ maxIterations **then**           **return** t       **end if**       tcut ← JuMP_MOA_Vector(Hcut; demand=true, demandScale)       **for** i in eachindex(B) **do**           t[B[i]]←tcut[i]       **end for**       count++  **end while**

## 7. Simulations and Results

In Brazil, ANEEL resolution defines that residential consumers are not charged for demand, and only those in the A group have this type of billing [12]. That way, the methodology presented in this topic would be enough for our local situation. Nevertheless, as discussed before, DR is a global concern and should be taken into account.

As mentioned before, the first simulation scenario is related to a reference house with 11 controllable loads which appliance set is familiar to us as they have been studied in previous related works [9,25,29,35]. Figure 8 illustrates the scheduling results in a cumulative or stacked load power. Figure 8a represents the house inhabitants’ preferences. Figure 8b,c show the geometric search and hierarchical results, both without demand constraints. At last, Figure 8d represents the schedule with demand constraint. The three least methodologies returned the same quantitative result, *videlicet*: hierarchical with DR, hierarchical with DR constraint and initialized with geometric search results, and the hybrid strategy. However, only the two purely based on the hierarchical algorithm were expected to return the same qualitative results. This result has occurred due to the small number of loads in this simulation set and because a significant amount of them were selected to run into hierarchical. In quantitative analysis, only the hybrid strategy has achieved better benchmarks, as hypothetically expected.

In Table 5, the values of comfort and normalized cost for reference house scenario can be read. Note that the best comfort values are realized by the methodology presented in this paper.

The results of the geometric search could be equal to hierarchical by adjusting the code to return the next inner point in the lower cost region, rather than the first one. All results achieved are compatible with cited previous works.

During experiments with the data from the first scenario, it was noted that the output results of the geometric search were significantly faster compared to those obtained by linear programming tools. Furthermore, less memory was also been allocated during the evaluation. To stand this result and verify its impact with larger sets of data, we proposed to generate eight sets of data with increasing size in $log_{10}$ scale, as described in Section 2. Each set of data has been submitted to five different scheduling procedures:

Geometric Search;Multi-objective hierarchical without demand constraint;Multi-objective hierarchical with demand constraint;Multi-objective hierarchical with demand constraint and initialized with Geofind solution;Hybrid algorithm.

The five scheduling results and the randomly generated expected time schedule can be examined in detail for each dataset in the link provided in Appendix A.

All benchmark results presented and discussed here have been run at least 25 times to ensure consistent outcomes. Most data has been collected through 100 or more executions under the same conditions and on the same computer (Intel Core i5-6200U 2.3 GHz with 8 GB of DDR4 Memory using Windows 10 Home and Julia 1.8). This iteration number has been considered sufficient as all results evaluated in each procedure consistently provided the same solution. Detailed histograms and all benchmark outputs can be found in the link provided in Appendix A.

Data in Table 6 shows the mean execution time for all random scenarios. Note that geometric search achieves results in microseconds while other solutions grow large reaching tens of seconds to display the results. Additionally, it is observed that the execution times collected for the hierarchical algorithm initialized with the output of geometric search is just slightly better than its non-initialized version. However, the hybrid proposed methodology has improved about 50% in comparison to hierarchical results. A similar discussion applies to memory estimate data presented in Table 7.

Figure 9 illustrates the benchmark results for time, while Figure 10 provides a close-up view to allow readers to discern the differences between two executions of the hierarchical algorithm and to appreciate the significant improvement achieved by the hybrid solution.

The next data presented here are the values for comfort (Table 8) and normalized cost (Table 9) evaluated for each load set applying all five scheduling methodologies. As occurred in scenario one, the geometric search has a better result in comfort metric than the hierarchical algorithm without DR restriction while achieving the goal to lower the energy cost due to constant tariff. The hybrid algorithm has also achieved better comfort metrics when compared to hierarchical results while all three algorithms have processed the DR constraint. For a better view of these results, the reader can refer to the graphics available in the link provided in Appendix A.

To finish our discussion about the results, Table 10 presents the dataset that has been processed by the hierarchical algorithm inside the hybrid solution and how many iterations it has executed to comply with the demand constraint.

Excluding the results presented in the first line of Table 10, after the scheduling has been processed by geometric search, an amount ranging from a third to a half of the loads in random datasets is agglutinated, causing a demand peak. So, through hybrid methodology, we have been able to reduce the amount of data processed by the LP tool, which explains the related improvement. Also note that the hybrid methodology has performed few search iterations, which also demonstrates the efficiency of this method. In the first case, the number of loads causing the demand peak where very close to the total quantity (9 out of 10).

That way, the search processing and running the hierarchical algorithm with reinforced constraint due to scale reduction results were similar to the original linear programming tool with 10 loads, so it is a valid effort.

## 8. Conclusions

Achieving simultaneous objectives of energy efficiency and comfort is not an easy task, as it represents an intricate trade-off between the need to reduce energy bill and stand for user preferences. The proposed solution performs fast optimization in a SH scenario whose ToU takes into account three tariff posts: off-peak, intermediate, and peak. That stated, the main goal of this work was to present a new methodology for scheduling home electric loads minimizing cost but without sacrificing inhabitants’ comfort rate.

To achieve the main goal, the methodology relies on defining tariff space, decomposing the time axis into multiple independent dimensions and establishing the geometric *locus* of a load. This *locus* models the behavior of an appliance as it progresses through either tariff space or time. It also emphasizes that a set of appliances can be represented as an independent set. Through explanation, examples of each definition have been provided and systematically explored.

A traditional optimization programming tool, like the hierarchical algorithm or any metaheuristic, begins computation from an initial value and explores the solution space to identify an optimal outcome. By employing the new concepts discussed in this paper, we determine the direct evaluation of the optimal solution to the given scheduling problem, eliminating the need for iterative exploration.

The benchmark results for processing time and memory usage, as presented in Table 6 and Table 7, respectively, illustrate a significant performance improvement compared to the solution evaluated by the LP multi-objective counterpart. Specifically, the processing time is approximately ten thousand times faster, and the memory usage is significantly reduced. These results underscore the efficiency and effectiveness of the proposed methodology in addressing the load scheduling problem without demand restriction.

The output results for comfort and cost, as presented in Table 8 and Table 9, highlight the reliability of proposed methodologies, as outputs for all methods are quite similar. It is anticipated that the new scheduler proposed in this work would achieve better results for the comfort metric. This expectation arises from the concept of the methodology, which aims to provide the shortest distance of movement in time necessary to reduce the energy bill. To ensure a fair comparison, the relevance factor μ, previously defined in Table 2, for all loads and in all simulations has been set to one. This choice prevents the diminishing of the comfort goal, as a relevance factor equal to zero for a load implies that shifting it beyond the expected time would cause no discomfort.

The presented methodology for scheduling appliances in a SH environment aligns with legal regulations in the Brazilian energy market. Moreover, it demonstrates the capability to solve large instances of this problem in less than a few milliseconds. Also, the reduction in memory usage by geometric search holds potential for the practical implementation of this solution in low-cost embedded systems. This advancement could enhance HEMS soon and contribute to popularizing this kind of equipment.

Besides optimization, achieving energy bill savings constrained by flattening demand is important as it contributes to reducing investments of energy suppliers in the distribution network. This helps decrease the constant need to expand the system to maintain the availability of ever-increasing energy consumption [9], while the current state of geometric search is not able yet to directly relate to demand restrictions, the present work has also tested two methodologies to enhance the geometric search methodology so it could also comply with demand response restriction problems.

The proposed hybrid algorithm achieved superior benchmark results compared to its linear programming counterpart, with improvements of at least 40% for larger load sets. However, better ways to include demand constraints should be also included in future works. Some other hypotheses using Game Theory [53] and Topological Algebra are currently under study.

Time itself is a substantially complex subject, and this work may have started to open a window that will allow us to explore its properties even further. The decomposition of time into a geometric space is a new methodology that is far from reaching its full potential. Future research should include extrapolation of tariff space for hour-based tariffs or even continuous ones. The works proposed by [59,60] seem to be a key path to continue this discussion.

## Figures and Tables

**Figure 1 sensors-24-01875-f001:**
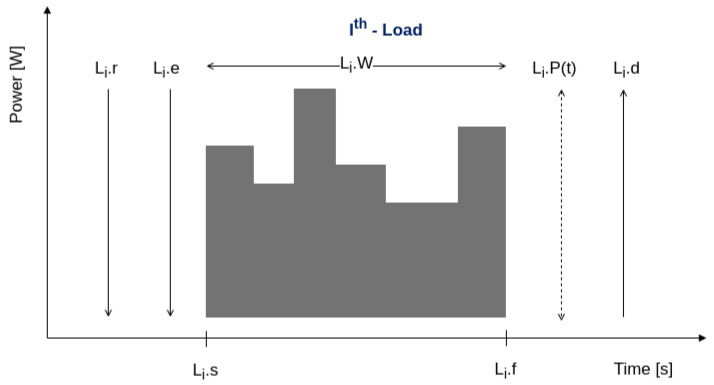
Load model and timing parameters.

**Figure 2 sensors-24-01875-f002:**
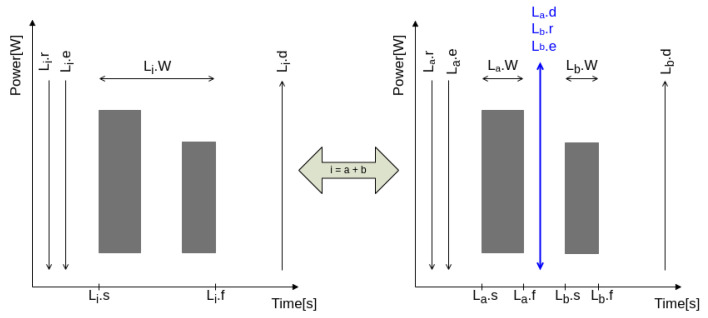
Multiple stage load to multiple simple loads.

**Figure 3 sensors-24-01875-f003:**
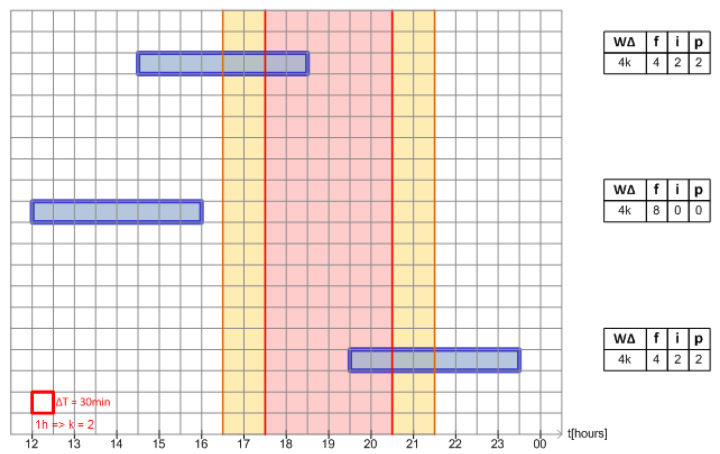
Three different startup times for a 4 h load around intermediate and peak post tariffs.

**Figure 4 sensors-24-01875-f004:**
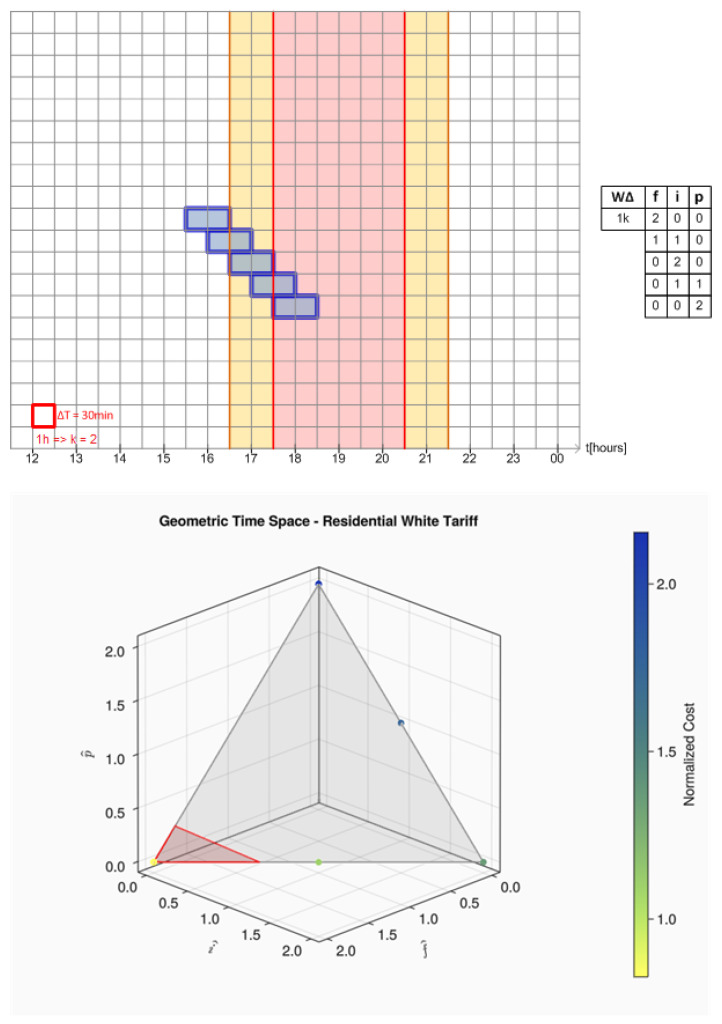
All startup times for a 1 h load crossing intermediate and peak post tariffs.

**Figure 5 sensors-24-01875-f005:**
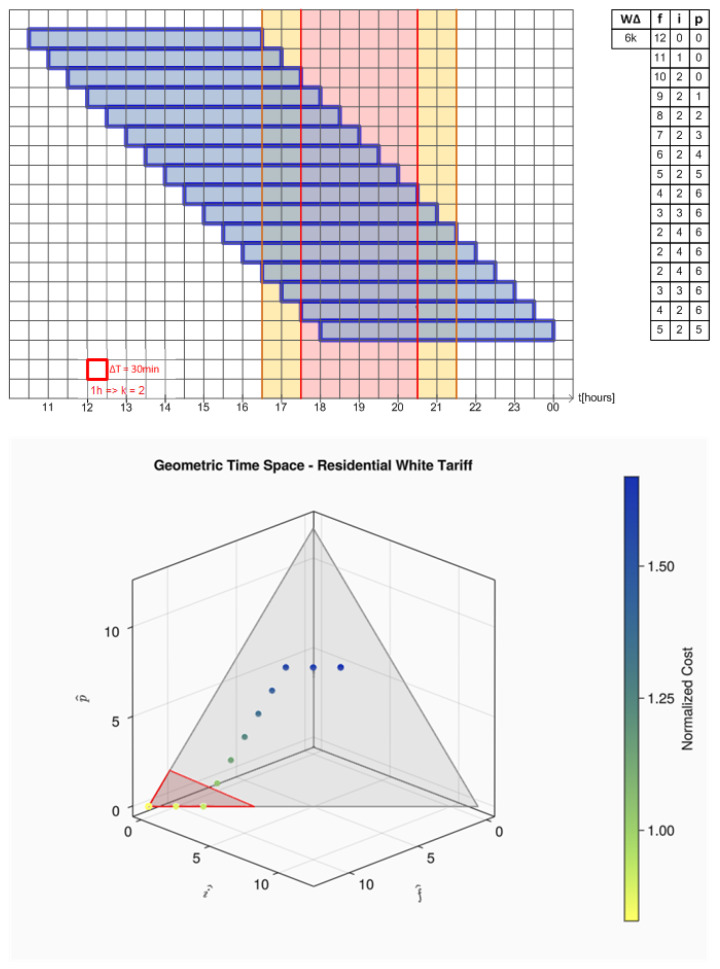
Example of startup times for a 6 h load crossing intermediate and peak post tariffs.

**Figure 6 sensors-24-01875-f006:**
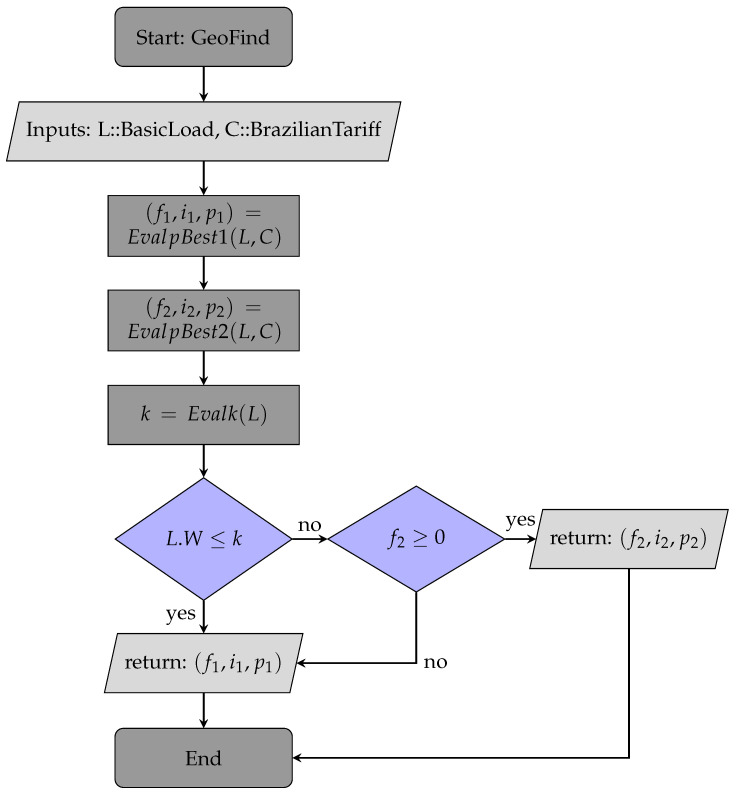
Geometric search flowchart.

**Figure 7 sensors-24-01875-f007:**
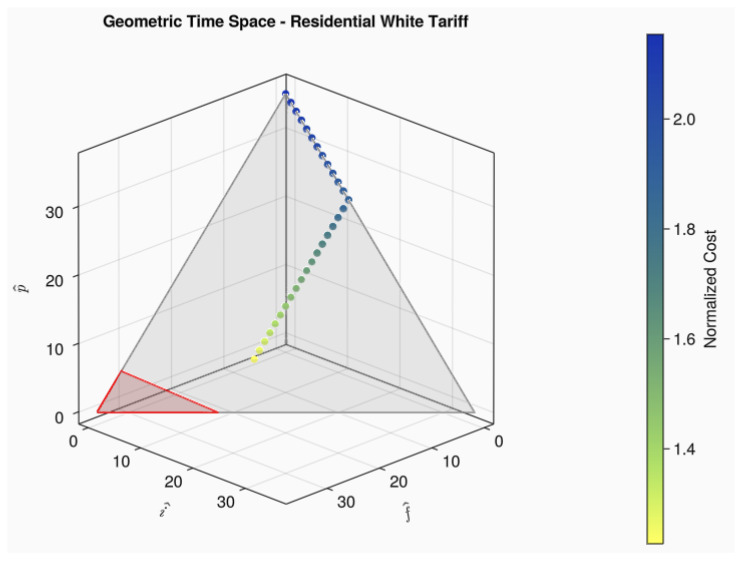
Example of a defective load into tariff space time.

**Figure 8 sensors-24-01875-f008:**
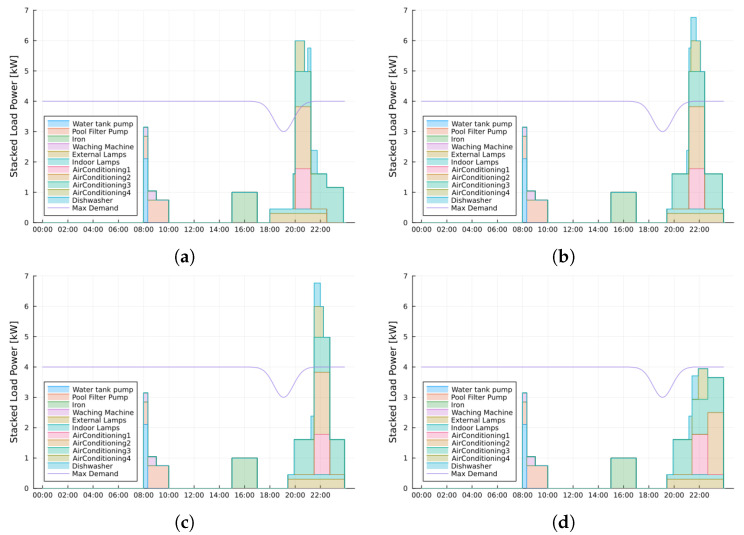
Reference house loads user schedule and output results. (**a**) Expected user defined schedule, maximum comfort, (**b**) Geofind Algorithm output, (**c**) Hierarchical without demand restriction, (**d**) Hierarchical with demand restriction and Hybrid Strategy.

**Figure 9 sensors-24-01875-f009:**
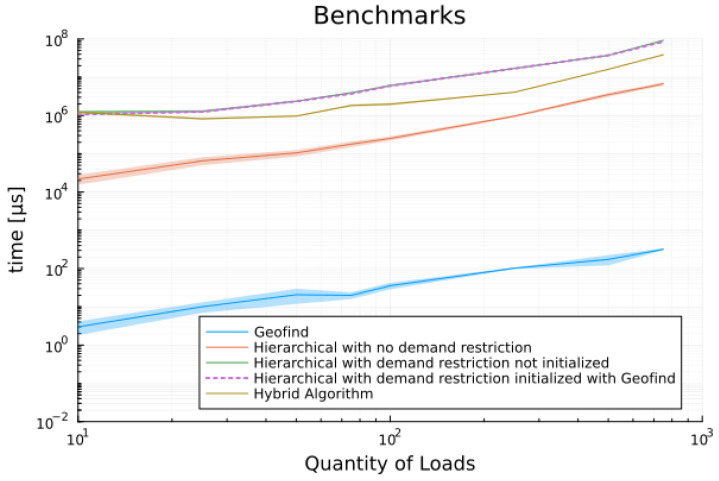
Benchmarks results for execution time.

**Figure 10 sensors-24-01875-f010:**
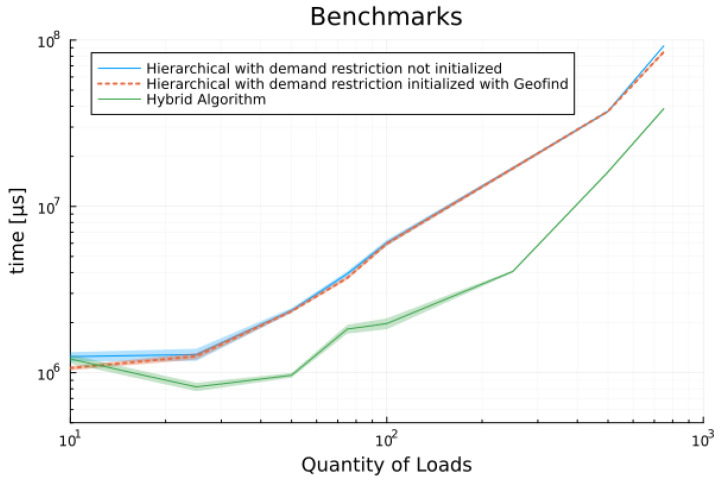
Benchmarks results for execution time, only algorithms with demand response constraint.

**Table 1 sensors-24-01875-t001:** Comparison between conventional tariff and white tariff over time.

Period	Tariff Post	$T_{C}$ (BRL/kWh)	White Tariff (BRL/kWh)
00:00 to 16:30	Off-peak	0.74373	0.62124
16:30 to 17:30	Intermediate	0.74373	1.03901
17:30 to 20:30	Peak	0.74373	1.63527
20:30 to 21:30	Intermediate	0.74373	1.03901
21:30 to 00:00	Off-peak	0.74373	0.62124

**Table 2 sensors-24-01875-t002:** List of symbols that define parameters of a *BasicLoad* structure in Julia language for a load L.

Id	Description
$L_{i}.r$	release time of the *i*th load
$L_{i}.e$	expected time of the *i*th load
$L_{i}.d$	deadline instant of the *i*th load
$L_{i}.s$	range with all possible start times of the *i*th load, $s_{i}\geq r_{i}$
$L_{i}.f$	finishing time of the *i*th load, $f_{i}=s_{i}+W\leq d_{i}$
Δt	discrete step in which a load could move through time, common to all loads in set
$L_{i}.W$	width of the *i*th load, usually measured in minutes
$L_{i}.W\Delta$	discrete load width
$P_{i}\left(t\right)$	behavior of the *i*th load through time
$\bar{P_{i}}$	average power of the *i*th load
$\hat{P_{i}}$	peak power of the *i*th load
$L_{i}.\mu$	relevance of the *i*th load, ∈[0,1]
*k*	proportion of discrete time related to 1 h, $k=\left(\frac{60}{\Delta t}\right)$

**Table 3 sensors-24-01875-t003:** Reference loads in an actual residence.

ID	Load	Cycles	Δt(min)	$\bar{P}\left[\mathrm{kW}\right]$	$\hat{P}\left[\mathrm{kW}\right]$	Expected Time	Release Time	Deadline Time	$\mu_{i}$
1	Water tank pump	1	20	2	3	8 h	7 h	17 h	1.0
2	Pool filter pump	1	120	0.75	1.5	8 h	7 h	17 h	1.0
3	Iron	1	120	1	1.2	16 h	14 h	17 h	1.0
4	Washing machine	8	10 10 4 6 2 2 2 7	0.13 0.51 0.30, 0.26 0.15 ⋯ 0.22	0.70 0.50 0.30 0.26 0.15 ⋯ 0.30	8 h	7 h	17h	1.0
5	External lamps	1	270	0.3	0.3	18 h	17 h	24 h	1.0
6	Indoor lamps	1	270	0.15	0.3	18 h	17 h	23 h	1.0
7	Air Conditioning 1	14	[10 5 5 ⋯ 5]	[1.3 ⋯ 1.3]	[1.7 1.3 ⋯ 1.3]	16 h	15h	24 h	1.0
8	Air Conditioning 2	7	[30 20 5 ⋯ 5]	[2 ⋯ 2]	[2.1 ⋯ 2.1]	20 h	17 h	24 h	1.0
9	Air Conditioning 3	1	240	1.1	1.2	20 h	17 h	24 h	1.0
10	Air Conditioning 4	7	[10 10 5⋯ 5]	[0,9 ⋯ 0.9]	[1,1 ⋯ 1.1]	20 h	17 h	24 h	1.0
11	Dishwasher	5	5, 10, 15, 5, 10	0.03, 1.76, 0.03, 1.76, 0.03	0.03, 1.76, 0.03, 1.76, 0.03	21 h	18 h	22 h	1.0

**Table 4 sensors-24-01875-t004:** Example of a load with defective geometric *locus*.

L.r	L.e	L.d	L.Δt	L.W
16 h	18 h	23 h	5 min	3 h

**Table 5 sensors-24-01875-t005:** Mean comfort and cost for reference house appliances.

Algorithm	Comfort	Cost
Expected user time	1.000	1.311
Geofind	**0.873**	1.021
Hierarchical without DR	0.845	**0.965**
Hierarchical with DR	0.813	0.974
Hierarchical with DR ^a^	0.813	0.974
Hybrid algorithm	0.813	0.974

^a^ Initialized with geometric search results.

**Table 6 sensors-24-01875-t006:** Mean execution time benchmarks for random loads scenario.

Random Loads	Geofind without DR	Hierarchical without DR	Hierarchical with DR	Hierarchical with DR ^a^	Hybrid Algorithm
10	3.005 μs	0.022 s	1.249 s	1.066 s	1.208 s
25	10.074 μs	0.065 s	1.285 s	1.256 s	0.822 s
50	20.748 μs	0.105 s	2.370 s	2.343 s	0.966 s
75	19.975 μs	0.179 s	3.933 s	3.715 s	1.829 s
100	35.647 μs	0.252 s	6.055 s	5.968 s	1.975 s
250	102.471 μs	0.962 s	16.864 s	16.919 s	4.061 s
500	173.276 μs	3.509 s	37.316 s	37.328 s	16.124 s
750	319.320 μs	6.699 s	92.339 s	84.897 s	38.664 s

^a^ Initialized with geometric search results.

**Table 7 sensors-24-01875-t007:** Memory estimate benchmarks for random loads scenario.

Random Loads	Geofind without DR	Hierarchical without DR	Hierarchical with DR	Hierarchical with DR ^a^	Hybrid Algorithm
10	1.17 KiB	5.84 MiB	37.84 MiB	37.84 MiB	36.53 MiB
25	3.66 KiB	19.09 MiB	92.67 MiB	92.64 MiB	60.82 MiB
50	5.45 KiB	40.51 MiB	180.14 MiB	180.07 MiB	82.99 MiB
75	7.50 KiB	72.37 MiB	273.62 MiB	273.53 MiB	153.42 MiB
100	15.59 KiB	123.39 MiB	421.92 MiB	421.79 MiB	161.99 MiB
250	31.16 KiB	508.57 MiB	1.21 GiB	1.21 GiB	303.17 MiB
500	68.56 KiB	1.72 GiB	3.13 GiB	3.13 GiB	1.22 GiB
750	97.38 KiB	3.67 GiB	5.93 GiB	5.93 GiB	2.67 GiB

^a^ Initialized with geometric search results. The ’i’ vowel in ’iB’ is short for integer.

**Table 8 sensors-24-01875-t008:** Mean normalized comfort ^a^ for random loads scenario.

Random Loads	Geofind without DR	Hierarchical without DR	Hierarchical with DR	Hierarchical with DR ^b^	Hybrid Algorithm
10	0.976	0.974	0.887	0.887	0.887
25	0.963	0.948	0.945	0.945	0.957
50	0.942	0.942	0.941	0.941	0.935
75	0.965	0.964	0.962	0.962	0.963
100	0.941	0.935	0.934	0.9341	0.945
250	0.968	0.959	0.958	0.958	0.967
500	0.947	0.934	0.937	0.937	0.944
750	0.954	0.940	0.941	0.941	0.946

^a^ Maximum possible value for comfort metric is one. ^b^ Initialized with geometric search results.

**Table 9 sensors-24-01875-t009:** Mean normalized cost for random loads scenario.

Random Loads	Geofind without DR	Hierarchical without DR	Hierarchical with DR	Hierarchical with DR ^a^	Hybrid Algorithm
10	0.871	0.828	0.843	0.843	0.843
25	0.923	0.893	0.893	0.893	0.914
50	0.859	0.833	0.839	0.839	0.839
75	0.873	0.852	0.852	0.852	0.873
100	0.910	0.863	0.863	0.863	0.907
250	0.881	0.847	0.847	0.847	0.881
500	0.897	0.854	0.858	0.858	0.885
750	0.885	0.846	0.848	0.848	0.862

^a^ Initialized with geometric search results.

**Table 10 sensors-24-01875-t010:** Hybrid algorithm running parameters for random loads scenario.

Random Loads	Loads Causing Demand Peak	Search Iterations
10	9	1
25	14	1
50	22	1
75	19	2
100	41	1
250	75	1
500	123	2
750	217	2

## Data Availability

Data is contained within the article.

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
