# Peer review of "A Scheduler for Smart Home Appliances Based on a Novel Concept of Tariff Space"

_sensors, 2024, doi:10.3390/s24061875_

Round 1

Reviewer 1 Report

Comments and Suggestions for Authors

 The presented paper dealing with smart home energy management and proposing to provide economic contribution by dealing with current tariffs is expected to attract interest to most of readers. The analytical approach is comprehensiveley described with mathematical modelling and different scenarios. I suggest authors to refine the equations to be understandable by readers, and to increase the trackability. It is possible to express anything with extended analytical expressions but it does not make sense if it has not any exchange on readers' side. As per my evaluation, although the expressions are trackable, it will not easy for most of readers. I have some comments for authors to provide minor revisions as follows;

      The simulation scenarios given under title 3.1 are described to follow house description given in ref. [29] and studies in [9, 25,35]. The house model and justification to select this model as a reference should be introduced and discussed in detail. The literature survey combines many different aspects and published papers. However, there is not a consensus among them to express the targeted study. Authors should present detailed and unified presentation of the literature survey including the recent smart home models and reference model cited at ref. [29]. Thus, readers will be able to understand the smart home models and accuracy of the proposed home model among others.

-          What does the “quantity of loads” mean on line 176? It is not clearly indicated how the reference real house model was tested in terms of load variation and which type of infrastructure used to acquire data from household devices. Please provide further description on these.

-          It is not clearly described that what SHC stands for, smart home controller or what else, please express at the first use of this abbreviation

-          There are some typos as on line 162, and many linked words in blue that do not bookmark to certain points but causes to motivate the reader to click links. I am not sure if these were intentionally located or mislinked. Please check this issue

Comments on the Quality of English Language

The English of paper is understandable and easy to track, but there are some typos I expressed above.

Reviewer 2 Report

Comments and Suggestions for Authors

Paper presents an interesting idea for dynamic tariff. The following suggestion/comments must be addressed to improve the quality of the paper.

1. There are number of typos and grammatical mistakes, a thorough proof read is required.

2. The literature review is not satisfactory and does not reflect the problem considered.

3. Figure 9 and 10 are not clear, why time has been considered in micro-seconds?

4. It is recommended to provide a detailed table to compare the user and supplier economical benefits of the proposed and existing techniques to prove the validity of concept.

5. Conclusion does not reflect the contribution of the paper.

6. Billing comparison with and without proposed technique for level of customers must be provided. 

7. How the comfort level of the customer has been considered, detailed analysis must be provided.

Comments on the Quality of English Language

There are number of grammar and spelling issues.

Reviewer 3 Report

Comments and Suggestions for Authors

This paper suffers from readability issues, primarily due to unclear explanations throughout. For instance, the distinction between controllable load and detectable load lacks clarity, and Figures 1 and 2 lack adequate elaboration. Additionally, numerous typos detract from the overall quality of the paper. The reviewer recommends that the authors carefully revise the paper and resubmit it for further review.

The topic is relevant. Many papers in the literature address similar problems, yet the author has not provided sufficient elaboration on how the problem settings or their proposed solution differ from existing work.

Comments on the Quality of English Language

·         Ensure that all figures and equations are thoroughly explained for clarity.

·         Leverage ChatGPT to enhance readability throughout the document.

Round 2

Reviewer 2 Report

Comments and Suggestions for Authors

I appreciate the efforts of the authors to improve the quality of the paper. It is an interesting paper introducing the concept of tariff space.